# A New Strategy for Bearing Health Assessment with a Dynamic Interval Prediction Model

**DOI:** 10.3390/s23187696

**Published:** 2023-09-06

**Authors:** Lingli Jiang, Heshan Sheng, Tongguang Yang, Hujiao Tang, Xuejun Li, Lianbin Gao

**Affiliations:** 1School of Mechanical and Electrical Engineering, Foshan University, Foshan 528000, China; linlyjiang@163.com (L.J.);; 2School of Mechanical Engineering and Automation, Northeastern University, Shenyang 110819, China; 3Wafangdian Bearing Co., Ltd., Wafangdian Bearing Industrial Park, Dalian 116300, China; 4Chengdu CRRC Electric Motor Co., Ltd., Chengdu 610500, China

**Keywords:** bearing, life prediction, dynamic interval data, state space equation, construction of health indicators

## Abstract

Bearing is the critical basic component of rotating machinery and its remaining life prediction is very important for mechanical equipment’s smooth and healthy operation. However, fast and accurate bearing life prediction has always been a difficult point in industry and academia. This paper proposes a new strategy for bearing health assessment based on a model-driven dynamic interval prediction model. Firstly, the mapping proportion algorithm is used to determine whether the measured data are in the degradation stage. After finding the starting point of prediction, the improved annealing algorithm is used to determine the shortest data interval that can be used for accurate prediction. Then, based on the bearing degradation curve and the information fusion inverse health index, the health index is obtained from 36 general indexes in the time domain and frequency domain through screening, fusion, and inversion. Finally, the state space equation is constructed based on the Paris-DSSM formula and the particle filter is used to iterate the state space equation parameters with the minimum interval data to construct the life prediction model. The proposed method is verified by XJTU-SY rolling bearing life data. The results show that the prediction accuracy of the proposed strategy for the remaining life of the bearing can reach more than 90%. It is verified that the improved simulated annealing algorithm selects limited interval data, reconstructs health indicators based on bearing degradation curve and information fusion, and updates the Paris-DSSM state space equation through the particle filter algorithm. The bearing life prediction model constructed on this basis is accurate and effective.

## 1. Introduction

Rolling bearing is one of the most important components in rotating machinery and its health directly affects the operation of the rotor system supported by it and even the whole machine [1]. Among the various faults of rotating machinery, the faults caused by rolling bearings are very common and the bearing life is a comprehensive reflection of the bearing quality [2,3]. Therefore, condition monitoring and life prediction of bearings and reasonable formulation of unit maintenance plans in advance can effectively improve the reliability of the entire mechanical equipment operation.

Life prediction has made great progress in the past 10 years. The main technical methods include data-driven prediction methods and model-driven prediction methods [4,5]. The data-driven prediction method uses the bearing performance degradation information in the monitoring data to establish a prediction model which can predict the remaining life of the bearing according to the current state and historical data of the bearing. For example, Wu et al. [6] constructed a long short-term memory (LSTM) autoencoder based on Hilbert transform to evaluate the trend of bearing degradation. Wang et al. [7] integrated time information into the convolutional neural network (CNN) and proposed a bearing remaining useful life (RUL) prediction method based on a recurrent convolutional neural network. Based on the data-driven model, Yang et al. [8] and Zhao et al. [9] have realized the bearing life prediction of small samples. However, the data-driven prediction method has certain requirements for the quality and diversity of data. How to reasonably select and process the monitoring data is an important challenge for establishing an accurate prediction model.

The method based on the mechanism model is mainly to construct a parametric mathematical model describing the degradation process of the equipment according to the failure mechanism, identify the parameters of the mathematical model by combining the design test data or empirical knowledge of the equipment, and then update the parameters of the mechanism model based on the condition monitoring data to realize the residual life prediction of the equipment. For example, Zhou N et al. [10] constructed the fatigue crack propagation model of the equipment and obtained the model parameters through experiments and simulation methods so as to simulate the crack propagation and RUL prediction of the two environments of the equipment. According to the physical characteristics of the product, Pugalenthi et al. [11] proposed an online prognostic model based on the particle filter and constructed a prediction framework to predict the striation expansion for the life prediction of the product. Zou L et al. [12] first proposed a fatigue life assessment method for press-fit axles that combines fretting wear, crack initiation, and propagation. The crack initiation was estimated by a mathematical physics model. The contour integral method and Paris law are used to predict crack propagation. Qu A et al. [13] conducted a comprehensive analysis of the mechanical and physical quantities of the product and studied the crack propagation law and residual life of the product. The model-driven prediction method establishes a physical or mathematical model to describe the degradation process of the object and combines the condition monitoring data to update the model parameters. Therefore, the model-driven prediction method can be well applied to the RUL prediction of the bearing.

The Paris-Erdogan model is a typical mechanism model that mainly explains the propagation and growth of micro-fatigue cracks in mechanical materials [14]. Since the degradation process of most machinery is directly or indirectly caused by micro-fatigue cracks, the Paris model is effective in describing the degradation process of machinery and is widely used in mechanical life prediction. The state space equation constructed by it can well describe the performance degradation evolution process of bearings. Kong W [15] et al. first determined the starting time of prediction and achieved good results in bearing RUL prediction. Liu S et al. [16] established a degradation model that integrates multiple degradation stages of bearings to predict the RUL distribution of different degradation stages. Zhang T et al. [17] extracted the average multi-scale morphological gradient power spectrum information entropy of rolling bearings and then used Hodrick Prescott trend filtering to process HI to construct a smooth HI. Qian et al. [18] and Liu et al. [19] used the optimization algorithm to update the physical model parameters so as to realize the bearing RUL prediction. The above model-driven prediction method needs to model the structure, material, and working conditions of the bearing. The above methods have achieved good prediction results for bearing RUL prediction. However, there are still some problems:

(1) The existing research can only provide a theoretical basis for predicting the starting time and it is difficult to find a basis for the selection of data length. In fact, the data length has a great influence on the prediction process. If the data are too long, it will bring computational redundancy. If the data are too short, it will affect the prediction accuracy. It is of great significance to study the shortest data to ensure prediction accuracy while minimizing computational redundancy.

(2) The existing research has proposed a lot of effective HI indicators, such as HI construction methods based on deep learning, but such methods rely on a large amount of data to train the model. Therefore, it is still worthy of continuous research to extract the sensitive features of the collected signals, fuse multiple extracted feature quantities, and fully use the information reflecting the bearing degradation process.

In view of the above problems, this paper proposes a new strategy for bearing life degradation prediction driven by a physical model based on bearing information of finite state monitoring. The main idea is to use the shortest interval data to predict the life of rolling bearings. The main innovations and contributions of this paper are as follows:

(1) A method for determining the shortest data interval for life prediction is proposed. Firstly, the mapping ratio algorithm is used to judge whether the measured data are in the degradation stage and find the starting point of prediction. Then, the improved annealing algorithm is used to increase the length of the data sample as the disturbance mechanism and the interval data are iterated to determine the shortest data interval that can be used for accurate prediction;

(2) A new method of HI index construction based on information fusion is proposed. Among the 36 initial time-domain and frequency-domain indicators, 12 sensitive indicators were screened according to monotonicity, trend, and prognosis. Furthermore, according to the bearing degradation curve, 12 sensitive indicators were fused to construct health indicators;

(3) A life prediction model based on Paris-DSSM is constructed. Based on the Paris-Erdogan model, the state space is constructed.

## 2. Bearing RUL Prediction Method Based on the Shortest Data Interval

### 2.1. Determination of the Shortest Data Interval Based on the Improved Annealing Algorithm

The whole life cycle of the bearing is divided into three stages: the health stage, degradation stage, and failure stage. As shown in Figure 1, it is an ideal bearing performance degradation curve. The health stage accounts for 80–90% of the complete life of the bearing [2]. In the health stage of the bearing, the operation of the bearing will produce some slight wear but this wear will not affect the normal operation of the bearing and the stability of the whole system. The degradation stage of the bearing is irreversible. When the cumulative damage of the bearing continues to increase, the cracks caused by bearing wear will gradually expand. As the bearing continues to operate, more stress concentration will occur which will aggravate the wear. Therefore, the degradation process at this stage is extremely obvious. In the failure stage of the bearing, the wear of the bearing continues to accumulate. When the cumulative damage reaches a certain extent, it will cause fatal damage to the bearing. The degradation of this stage shows an exponential growth [16].

The life prediction of the bearing should start from the degradation stage. The mapping proportional algorithm can be used to divide the health state of the bearing and find the prediction starting interval for the life prediction. The specific process is as follows.

Firstly, the smooth peak curve of the signal is calculated and the smooth peak curve is used as the mapping sample B of the mapping proportional algorithm.
(1)$$B_{\;}=\{b_{1}^{\;},b_{2}^{\;},\ldots ,b_{N_{\;}}^{\;}\}$$

Each element *b* is the point corresponding to the vibration amplitude of the time series and N is the length of the data.

The projection directions of the two mapping lines are $\mu_{1}$ and $\mu_{2}$ and the samples after projection are:(2)$${\left\{\begin{array}{c} Y=\mu_{1}B\\ X=\mu_{2}B \end{array}\right.}_{\;}$$

The sample mean $m_{i}$ and the within-class dispersion matrix S of the original sample space are:(3)$$m_{i}=\frac{1}{N_{\;}}\sum_{b_{j}^{\;}\in N_{\;}}B_{j}\;i=X,Y$$
(4)$$S=\sum_{b_{j}^{\;}\in B_{\;}}{\left(b_{j}-m_{i})(b_{j}-m_{i}\right)}^{T}$$

When the obtained mean value and intra-class dispersion meet the linear standard, the definition of the mapping proportional algorithm is calculated as follows:(5)$$F\left(B\right)=\frac{\left\|Y_{\;}\right\|}{\left\|X_{\;}\right\|}$$

The value of intra-class dispersion can reflect the fluctuation of data. The calculation formula is:

When $F(B)\leq \delta_{1}$ is the healthy stage, $\delta_{1}\leq F(B)\leq \delta_{2}$ is the degradation stage and $F(B)\geq \delta_{2}$ is the damage stage; the Formula (6) for $\delta_{1}$ and $\delta_{2}$ is as follows:(6)δ1=ϵ ℵ nfμ1τμ2δ2=ϵ ℵ nfμ130τμ2
where ϵ is the time factor, ℵ is the speed conversion factor, n is the speed, f is the sampling frequency, and τ is the time interval of data acquisition.

Through the above mapping ratio algorithm, the bearing can be judged whether it is in the degradation stage according to the small data and the starting point of life prediction can be found. However, for further life prediction, the length of data is very important. When data are too short the prediction accuracy is affected and when the data are too long computational redundancy is manifested. The paper proposes the determination of the minimum data length based on the improved annealing algorithm to ensure the prediction accuracy while minimizing the computational redundancy. The basic principles of this method are as follows:

The Simulated Annealing Algorithm (SAA) is a random search algorithm used to solve complex problems. Its purpose is to find the optimal solution through iteration, which is named after the physical heat treatment of materials. The traditional simulated annealing algorithm is at low temperature and searches into local optimum. The paper adopts the improved annealing algorithm. The disturbance mechanism in the algorithm is to increase the length of the data sample, dilute the disturbance process, and can adaptively accept the data samples after different data increments. In the search process, the current optimal solution is remembered and updated in time to reflect the best solution encountered in the search process to avoid the loss of the current optimal solution due to the execution probability acceptance link. The RUL prediction data interval is determined through the optimal solution update process. The improved annealing algorithm allows a certain probability to accept the inferior solution in the search process (the mapping ratio does not meet the solution of the degradation stage standard) so as to avoid falling into the local optimal solution. The initial data interval length is short and the probability of inferior solution decreases with the increase in data length until the final solution converges. The corresponding data interval length is the minimum RUL prediction interval.

In summary, the shortest data interval determination process that can achieve accurate life prediction is as follows:

(1) The signal is obtained and the smooth peak curve is extracted by signal processing;

(2) The initial measurement data sample is B; solve F(B) and judge whether the data are in the degradation stage. If not, return to step (1), if yes, turn to step (3);

(3) The collected samples are added to obtain a new sample set $B^{\prime }$ and the $F\left(B^{\prime }\right)$ after the sample changes is calculated;

(4) The function difference $∆F=F\left(B^{\prime }\right)-F(B)$ is calculated. If ∆F≥0, the new solution is received as the current solution; let $B=B^{\prime }$. If ∆F<0, the probability *p* is taken as the new solution, $p=e^{-\{F\left(B^{\prime }\right)-F(B)\}/F(K/N)}$; K is the proportional parameter which is related to the working condition and sampling frequency;

(5) Iterate steps (3)–(4) until the termination condition is satisfied and determine that the current data are the minimum length prediction data.

### 2.2. Health Index Inversion Based on the Bearing Degradation Curve and Information Fusion

The construction of health indicators is the key to accurate life prediction. It is difficult to accurately reflect the degradation process of bearings by a single time domain index or frequency domain index. This paper proposes to screen 12 sensitive indicators from 36 initial indicators. Furthermore, according to the bearing degradation curve, 12 sensitive indicators are fused to construct health indicators for life prediction.

There are 36 initial indicators, including 12 time-domain indicators, 12 frequency-domain indicators extracted after Fourier transform, and 12 frequency-domain indicators extracted after Hilbert transform. Time domain index is a feature extraction method based on time series data. It extracts some representative indexes to describe the running state of mechanical equipment by analyzing the change characteristics of signals in the time domain. The time domain statistical feature is the most basic characteristic parameter of rotating machinery, which is divided into two parts: dimensionless and dimensionless. The dimensionless parameter index is relatively easy to obtain and through the direct analysis of the original signal it is not affected by the bearing speed, bearing size, bearing, or other parameters. It is assumed that the collected acceleration signal is X=[x(1),…,x(n),…,x(N)] where N is the sampling length of the signal. Fourier transform interprets the random signal as a sine wave (processing stationary signal) signal with different frequencies orthogonal to each other. After Fourier transform, the spectrum is obtained. The collected acceleration signal is analyzed by the spectrum and the frequency domain data are S=[s(1),…,s(k),…,s(K)]. The extract 12 frequency domain features signals. In order to fully reflect the fault state information, a Hilbert transform is performed on the collected bearing fault signal. The Hilbert transform performs a phase transform on the frequency domain signal and then the inverse Fourier transform is performed to obtain the complex analytical signal as H=[h(1),…,h(k),…,h(K)] and 12 envelope spectrum characteristic signals are extracted.

The 12 sensitive indicators were obtained by screening 36 initial indicators based on monotonicity, trend, and prognosis and 12 sensitive indicators were screened out, as shown in Table 1.

The state space equation is a mathematical model used to describe the behavior of dynamic systems. The dynamic state space model is mainly composed of two equations: one is the state equation, which reflects the state of the dynamic system at a certain time point under the influence of input variables. This state describes the internal state of the system and is the essential change in the system; the second is the observation equation, which outputs the state of the system at a certain time point in some form, reflecting the change in the system with the change in input variables.
(7)$$t_{1}=\left(\frac{t}{T-1}-0.5\right)*2/k$$
(8)a=atanh (1/k)
(9)$$HI=\frac{\mathrm{atanh}\left(t_{1}\right)-\mathrm{atanh}\;(-\frac{1}{k})}{a-\mathrm{atanh}\;(-\frac{1}{k})}$$

In the formula, *t* is the eigenvalue time variable, *T* is the period of the eigenvalue, and *k* is the characteristic frequency.

### 2.3. Lifetime Prediction of a Particle Filter Based on the Paris-DSSM Formula State Space Equation

The state space equation is a mathematical model used to describe the behavior of dynamic systems. The dynamic state space model is mainly composed of two equations: one is the state equation, which reflects the state of the dynamic system at a particular time point under the influence of input variables. This state describes the internal state of the system and is the essential change in the system; the second is the observation equation, which outputs the state of the system at a particular time point in some form, reflecting the change in the system with the change in the input variable [20].

The state space model is usually defined as follows:(10)$$\mathrm{State}\;\mathrm{equation}:\;\;\;x_{t}=f_{t}(x_{t-1},\omega_{t})t_{1}=\left(\frac{t}{T-1}-0.5\right)*2/k$$
(11)$$\mathrm{Observation}\;\mathrm{equation}:\;\;\;\;\;\;y_{t}=h_{t}(x_{t},v_{t})$$
where $f_{t}$ and $h_{t}$ are both nonlinear functions, $w_{t}$ is the process noise, and $v_{t}$ is the observed noise; they are independent and unrelated noise sequences.

Due to the difficulty of developing the physical model of the bearing and the complexity of establishing the model, the paper predicts the bearing degradation by improving and deriving the Paris formula. The crack propagation formula is usually [14,21]:(12)$$\frac{dl}{dM}=C{\left(\Delta k\right)}^{\gamma }$$

In the formula, *l* is the crack length, *M* is the number of stress cycles, *C* and *γ* are the relevant parameters of the material, and ∆k is the range of stress intensity factor. The crack length *l* is difficult to measure in the working process of the bearing. Now, the Paris model is transformed as follows:(13)$$\alpha =C\varepsilon^{\gamma },\beta =\gamma /2,\frac{dl}{dM}=\alpha l^{\beta }$$

The state space model is set as:(14)$$\left\{\begin{array}{c} l_{k}=l_{k-1}+\alpha_{k-1}l_{k-1}^{\beta }\Delta t_{k}\\ \alpha_{k}=\alpha_{k-1}\\ s_{k}=x_{k}+v_{k} \end{array}\right.$$
where $l_{k}$ is the health indicators in k state; β is a constant parameter, which obeys the normal distribution $N(0,\sigma_{v}^{2})$; and $s_{k}$ is the predicted value of the final state. $\alpha_{k-1}$ obeys the normal random distribution $N(\mu_{a},\sigma_{a}^{2})$, β is a constant parameter, and $v_{k}$ is the observed noise vector value at time t and obeys the normal distribution $N(0,\sigma_{v}^{2})$.

In this paper, the particle filter algorithm is used to implement iterative optimization. The specific process is as follows:

(1) Initialize the particle set. The initialization step of the particle filter algorithm involves generating a set of particles that represent the possible state of the system. These particles are sampled from a prior distribution that captures our knowledge of the system before any measurement is performed. The choice of prior distribution depends on the nature of the system to be modeled. Let k = 0; the initial value p(x_0_) of the prior probability distribution is used to generate the sampling particles ${\left\{x_{0}^{i}\right\}}_{i=1}^{N},\omega_{0}^{i}=1/N$;

(2) Weight calculation and update. The weight update is usually carried out using Bayesian rules which update the particle weight according to the likelihood of the observation data of the given particle state and the prior probability of the particle state [22].
(15)$$\omega_{k}^{\left(i\right)}=\omega_{k-1}^{\left(i\right)}\frac{p(y_{k}|{\overset{\sim }{x}}_{k}^{\left(i\right)})p({\overset{\sim }{x}}_{k}^{\left(i\right)}|{\overset{\sim }{x}}_{k-1}^{\left(i\right)})}{q({\overset{\sim }{x}}_{k}^{\left(i\right)}|{\overset{\sim }{x}}_{0:k}^{\left(i\right)},y_{1:k})}$$
(16)$$\omega_{k}^{\left(i\right)}=\frac{\omega_{k}^{\left(i\right)}}{\sum_{j=1}^{N}\omega_{k}^{j}}$$
where $p(y_{k}|{\overset{\sim }{x}}_{k}^{\;})$ is the observed likelihood probability density, $p({\overset{\sim }{x}}_{k}^{\;}|{\overset{\sim }{x}}_{k-1}^{\;})$ is the state transition probability density, and $q({\overset{\sim }{x}}_{k}^{\;}|{\overset{\sim }{x}}_{0:k}^{\;},y_{1:k})$ is the importance function [23];

(3) Resampling. According to the weight of the particles, a new set of particles is selected from the current particle set. The main idea is to increase the particles with larger weights and reduce the particles with smaller weights. The new particle set can better represent the posterior distribution of the system state. The judgment criterion $N_{eff}=\frac{1}{\sum_{i=1}^{N}\;{\left(w_{k}^{\left(i\right)}\right)}^{2}}$ is selected to determine the number of effective particles and the number is used to determine whether resampling is performed. If $N_{eff}\geq N_{\mathrm{thres}\;}$, then $x_{0:k}^{\left(i\right)}=x_{0:k}^{\left(i\right)}$ and $\omega_{k}^{\left(i\right)}=\omega_{k}^{\left(i\right)}$, otherwise: $\kappa_{i}=l:x_{0:k}^{\left(i\right)}=x_{0:k}^{\left(\kappa_{i}\right)}$ and $\omega_{k}^{\left(i\right)}=1/N$, $\omega_{k}^{\left(i\right)}$ is the importance weight [24];

(4) State prediction. According to the state equation, the state value of the next moment is predicted;
(17)$$x_{0:k}^{est}=\sum_{i=1}^{N}x_{0:k}^{\left(i\right)}w_{k}^{\left(i\right)}$$

(5) Particle update: Use the predicted state value to update the state value of each particle in the particle set to obtain a new particle set;

(6) Repeat the above steps until the required state value is estimated or the termination condition is reached;

(7) The state space model is established, the HI is brought into the model, and the particle filter algorithm is used to estimate the model parameters and states. That is, the state change in the system is calculated according to the existing observation data and the model so as to predict its future operating state;

(8) The future state is predicted at the predicted time point. When the predicted state reaches the set failure alarm threshold, the time to reach the value is calculated as the remaining service life.

The process of bearing the RUL prediction method based on the shortest data in-terval is shown in Figure 2. Firstly, the minimum prediction data is determined, and then the HI index of the determined minimum prediction data is extracted. Finally, the state space model is constructed to predict the remaining life of RUL.

## 3. Experiment and Analysis

### 3.1. Data Acquisition

The verification of the proposed method is carried out on the XJTU-SY data set acquisition experimental platform [25] and the experimental platform is shown in Figure 3. The platform is composed of an AC motor, motor speed controller, shaft, support bearing, hydraulic loading system, and test bearing. It can carry out accelerated life tests of various rolling bearings or sliding bearings under different working conditions and obtain the whole life cycle monitoring data of test bearings. The paper verifies that the data are derived from the test bearing LDK UER204 rolling bearing. The data set is derived from the accelerated life test of the bearing under different operating conditions. The measured data include three working conditions, as shown in Table 2. The vibration acceleration signal is collected. The sampling frequency is 25.6 kHz, the sampling interval is 1min, and the single sampling time is 1.28 s.

### 3.2. Determination of the Shortest Data Interval for Prediction

It is assumed that the data are collected at the time of *t1* and the smooth peak curve is obtained for the vibration signal after time *t1*. As shown in Figure 4a, firstly, according to the working condition parameters, $\delta_{1}=5.323\times 10^{-5}$, $\delta_{2}=1.971\times 10^{-4}$ is obtained as well as the mapping ratio value $F\left(B\right)=\frac{\left\|Y_{\;}\right\|}{\left\|X_{\;}\right\|}=6.511*10^{-5}$ and $\delta_{2}>F\left(B\right)>\delta_{1}$. It is judged that the time has entered the degradation stage. Using the improved simulated annealing algorithm, the collected vibration signals are continuously increased for iterative calculation. In the iterative calculation process, when the probability of the inferior solution is close to 0, the vibration signal involved in the calculation is the shortest data interval that can achieve accurate RUL prediction, as shown in Figure 4b.

### 3.3. Construction of Health Indicators

The data Bearing1_1 are selected and a total of 36 feature indexes in the time domain and frequency domain are extracted. According to the monotonicity, trend, and prognosis of the features, a total of 12 features of F1–F12 shown in Table 1 are selected, as shown in Figure 5. These 12 features are fused with reference to the ‘bearing degradation curve’ and normalized to construct HI. The HI curve is shown in Figure 6.

### 3.4. Analysis of Life Prediction Results

The initial values of the parameters of the dynamic state space equation and the predicted data parameters are set; the state of the state space model will be continuously updated and iterated according to the observation information. The experimental verification of 13 sets of bearing data in Table 2 is carried out. The HI is input into the state space model and the model parameters are updated by the particle filter method. The parameter update process of each bearing data is shown in Figure 7. It can be observed that the parameters gradually converge to a constant value over time.

The life prediction process of each piece of data based on the Paris-DSSM formula state space equation is shown in Figure 8. It can be observed from the results in the figure that on the limited interval data selected by the improved simulated annealing algorithm, the prediction method of updating the model parameters by the particle filter algorithm not only has high prediction accuracy for the dynamic interval prediction model but also can achieve accurate prediction of the evolution law of the bearing performance degradation state.

From Figure 8, it can be observed that the method proposed in this paper is more accurate to reflect the degradation process of the bearing.

The prediction accuracy is measured according to the percentage error. The calculation error method is shown in Formula (18), $L_{a}$ is the actual RUL, $L_{p}$ is the predicted RUL, and the life prediction results are shown in Table 3.
(18)$$Er=|\frac{L_{a}-L_{p}}{L_{a}}|\times 100\%$$

According to the experimental results, this method has a good prediction result for the life of the bearing by using limited interval data and the prediction accuracy of half of the bearings reaches more than 90%. Among them, Bearing1_3, Bearing2_3, and Bearing2_5 have a sudden change in the later stage of life, resulting in a large prediction error.

In order to verify the superiority of the improved annealing algorithm compared with the traditional annealing algorithm, the results of life prediction and all data participation in life prediction are further compared with the shortest data interval determined by the traditional annealing algorithm. The results are shown in Figure 9.

From Figure 9, it can be observed that the convergence effect of the interval data selected by the improved simulated annealing algorithm on the updating of the parameters of the state space equation is similar to that of all the data participating in the updating; the final prediction accuracy is also similar: 88.7% and 89.5%, respectively. However, the traditional simulated annealing algorithm is easy to enter the local optimum in the process of updating iterations. Therefore, the iterative data interval is short and cannot meet the convergence of the parameters of the state space equation and the prediction accuracy is relatively low.

## 4. Conclusions

Due to the influence of many factors, such as material fatigue, friction and wear, and lubrication state, the degradation mechanism of rolling bearings is very complicated. At the same time, it is very difficult to obtain data on the whole life cycle of rolling bearings. In this paper, a life prediction method based on a small amount of condition monitoring information and the bearing degradation physical model is proposed. The main conclusions are as follows:

(1) The mapping ratio algorithm is used to determine whether the measured data are in the degradation stage and find the starting point of prediction. Then, the improved annealing algorithm is used to increase the length of the data sample as the disturbance mechanism and the interval data is iterated to determine the shortest data interval that can be used for accurate prediction. It is an effective method for determining the minimum data interval;

(2) Among the 36 initial time domain and frequency domain indexes, 12 sensitive indexes are selected according to monotonicity, trend, and prognosis. Furthermore, according to the bearing degradation curve, the 12 sensitive indexes are fused to reconstruct the health index, which can accurately reflect the bearing performance degradation;

(3) The state space equation is constructed based on the Paris-Erdogan model. The particle swarm optimization algorithm is used to iteratively update the parameters by using the data in the shortest data interval. The prediction model can be accurately used for bearing life prediction. The prediction accuracy of XJTU-SY rolling bearing data is above 90%.

Subsequent work plan/work to be studied:

The next step will consider how to find enough life prediction data and deal with the rule prediction problem in the big data environment. Nowadays, with the iterative development of mechanical equipment, the number of mechanical equipment is increasing and the number of sensors is also increasing. At the same time, with the advent of GPT, big data bring excellent opportunities for bearing life prediction but also significant challenges. For example, how to find valuable data among massive data [26,27].

The construction of the state space model in this paper is how to characterize the common faults of the bearing. The RUL prediction of the bearing combined with the fault diagnosis can not only predict the life of the bearing but also diagnose the fault type of the bearing [28,29].

With regard to how to improve the generalization ability of bearing health state division, according to the data of different working conditions, the apparent characteristics of stage division are extracted to make data before RUL prediction and improve the efficiency of bearing RUL prediction [30,31].

## Figures and Tables

**Figure 1 sensors-23-07696-f001:**
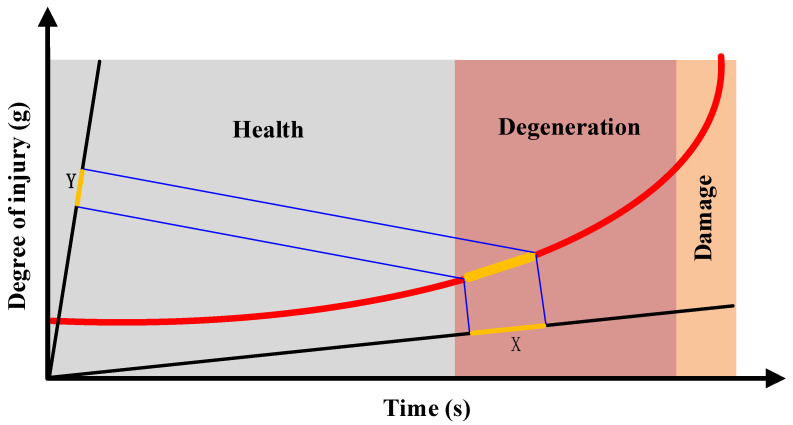
Bearing degradation curve.

**Figure 2 sensors-23-07696-f002:**
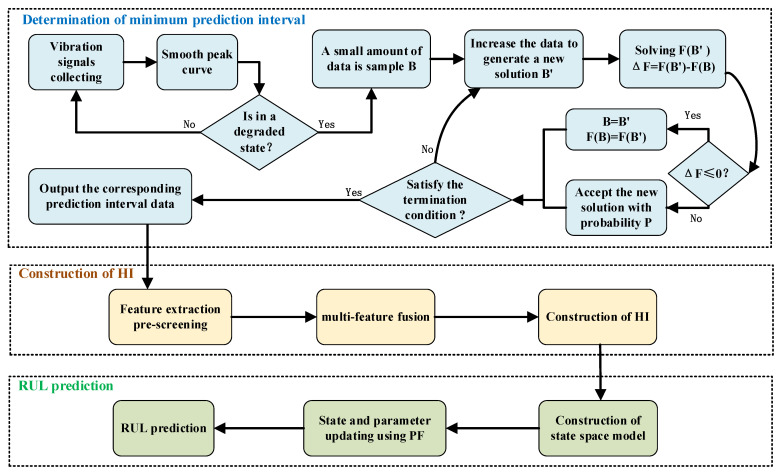
Method flow chart.

**Figure 3 sensors-23-07696-f003:**
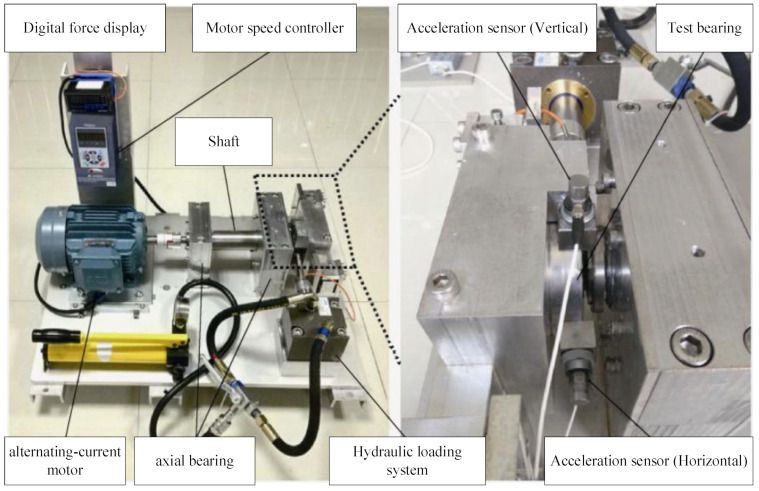
Bearing accelerated life test bench.

**Figure 4 sensors-23-07696-f004:**
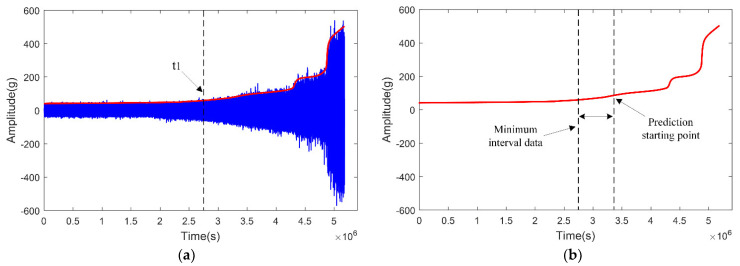
Determination of the shortest data interval for prediction. (**a**) Extract smooth peak curve, and (**b**) Determination of minimum prediction interval.

**Figure 5 sensors-23-07696-f005:**
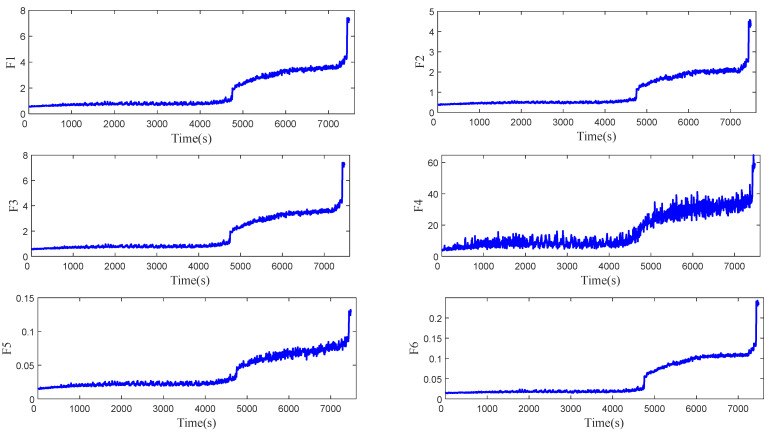
Curves of 12 sensitive characteristic indexes.

**Figure 6 sensors-23-07696-f006:**
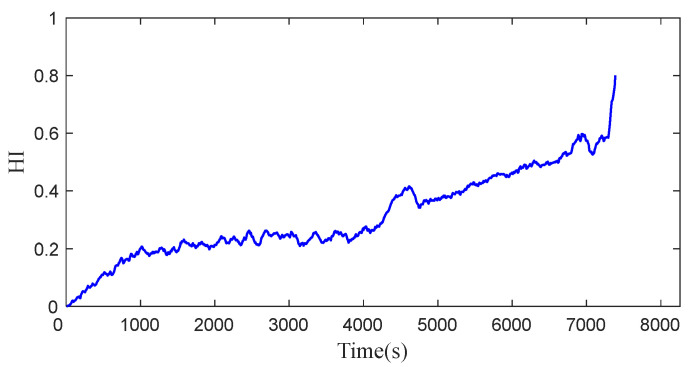
HI obtained by fusion.

**Figure 7 sensors-23-07696-f007:**
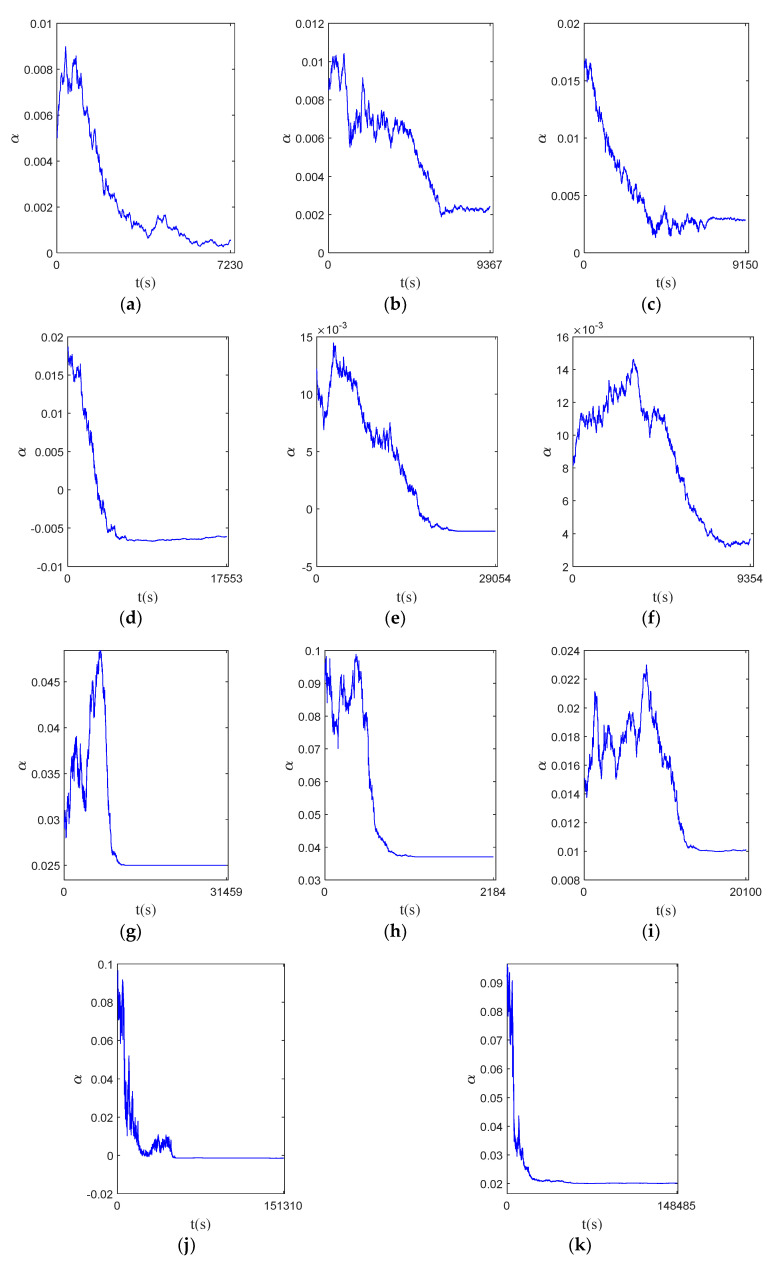
Parameter update process. (**a**) bearing1_1; (**b**) bearing1_2; (**c**) bearing1_3; (**d**) bearing1_5; (**e**) bearing2_1; (**f**) bearing2_2; (**g**) bearing2_3; (**h**) bearing2_4; (**i**) bearing2_5; (**j**) bearing3_1; (**k**) bearing3_2; (**l**) bearing3_4; and (**m**) bearing3_5.

**Figure 8 sensors-23-07696-f008:**
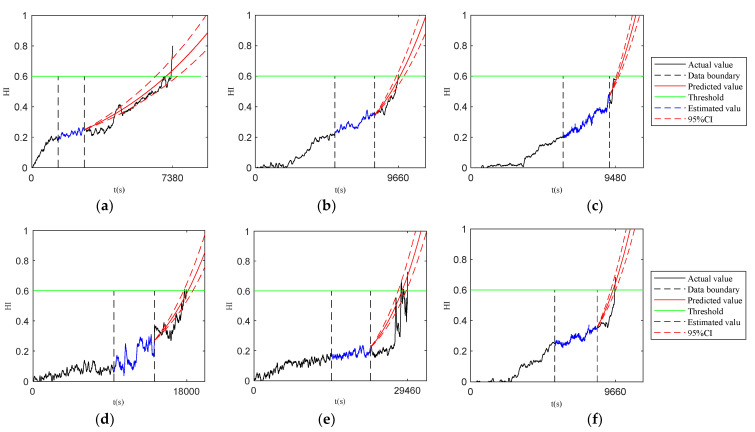
Life prediction experiments under different working conditions. (**a**) bearing1_1; (**b**) bearing1_2; (**c**) bearing1_3; (**d**) bearing1_5; (**e**) bearing2_1; (**f**) bearing2_2; (**g**) bearing2_3; (**h**) bearing2_4; (**i**) bearing2_5; (**j**) bearing3_1; (**k**) bearing3_2; (**l**) bearing3_4; and (**m**) bearing3_5.

**Figure 9 sensors-23-07696-f009:**
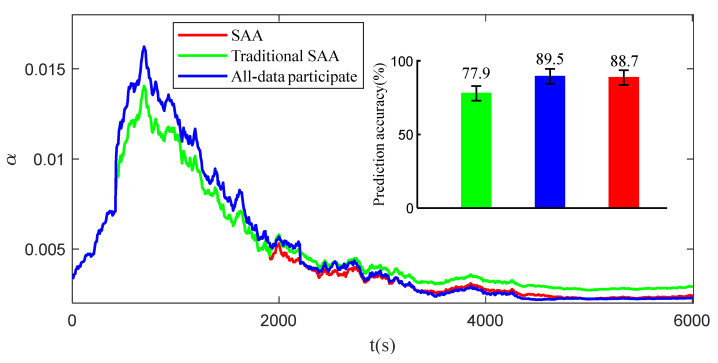
Life prediction experiments under different working conditions.

**Table 1 sensors-23-07696-t001:** Extract feature index formula summary.

12 Feature Extraction Formulas		
$F_{1}=\sqrt{\frac{1}{N-1}\sum_{n=1}^{N}{\left[x(n)-\bar{x}\right]}^{2}}$	$F_{2}=\sqrt{\frac{1}{K-1}\sum_{k=1}^{K}{\left[s(k)-F_{12}\right]}^{2}}$	$F_{3}=\sqrt{\frac{1}{N}\sum_{n=1}^{N}x^{2}(n)}$
$F_{4}=\max(x)-\min(x)$	$F_{5}=\frac{\sum_{k=1}^{K}{\left(f_{k}-F_{16}\right)}^{3}s(k)}{(K-1)F_{17}^{3}}$	$F_{6}=\frac{\sum_{k=1}^{K}{\left(f_{k}-F_{16}\right)}^{4}s(k)}{(K-1)F_{17}^{4}}$
$F_{7}=\frac{\sum_{k=1}^{K}f_{k}^{2}s(k)}{\sqrt{\sum_{k=1}^{K}s(k)\sum_{k=1}^{K}f_{k}^{4}s(k)}}$	$F_{8}=\sqrt{\frac{\sum_{k=1}^{K}f_{k}^{2}s(k)}{\sum_{k=1}^{K}s(k)}}$	$F_{9}=\frac{1}{K}\sum_{k=1}^{K}s(k)$
$F_{10}=\frac{1}{K}\sum_{k=1}^{K}h(k)$	$F_{11}=\sqrt{\frac{\sum_{k=1}^{K}f_{k}^{2}h(k)}{\sum_{k=1}^{K}h(k)}}$	$F_{12}=\frac{\sum_{k=1}^{K}f_{k}^{2}h(k)}{\sqrt{h(k)\sum_{k=1}^{K}f_{k}^{4}h(k)}}$

**Table 2 sensors-23-07696-t002:** XJTU-SY full life cycle bearing data set (part).

	Mode 1 (Load 13 KN, Speed 2100 RPM)	Mode 2 (Load 11 KN, Speed 2250 RPM)	Mode 3 (Load 10 KN, Speed 2400 RPM)
Data	Bearing1_1 Bearing1_2 Bearing1_3 Bearing1_5	Bearing2_1 Bearing2_2 Bearing2_3 Bearing2_4 Bearing2_5	Bearing3_1 Bearing3_2 Bearing3_4 Bearing3_5

**Table 3 sensors-23-07696-t003:** Initial value setting of parameters.

Bearing	Prediction Time Period/s	Actual RUL/s	Prediction RUL/s	Error/%
Bearing1_1	1376–2753	4513	4314	4.41
Bearing1_2	5944–8056	1586	1706	−7.57
Bearing1_3	6055–9084	346	529	−52.89
Bearing1_5	9460–14,209	3530	3968	−12.41
Bearing2_1	14,993–22,350	5825	6049	−3.85
Bearing2_2	5609–8390	1177	1020	13.34
Bearing2_3	19,097–28,055	5145	7763	−50.88
Bearing2_5	915–1919	687	768	−11.79
Bearing3_1	11,127–15,006	3885	5457	−40.46
Bearing3_2	83,896–116,685	7165	6982	2.55
Bearing3_3	73,243–106,659	38,545	36,493	5.32
Bearing3_5	43,870–63,901	24,608	23,711	3.65

## Data Availability

Not applicable.

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
