# Peer review of "A New Strategy for Bearing Health Assessment with a Dynamic Interval Prediction Model"

_sensors, 2023, doi:10.3390/s23187696_

Round 1

Reviewer 1 Report

The references should be adjusted in accordance with journal's standard ([reference_number] except for (Surname1 (et al.))

Author Response

We genuinely appreciate your input and believe that these modifications address the concerns raised. We are grateful for the opportunity to enhance the quality and clarity of our research.

Reviewer 2 Report

Overall structure of the manuscript is very poorly organized, Introduction do not have sufficient studies to support the theme of manuscript, table one is useless it should be integrated in text form in introduction. Material and method section is also poorly structured, Lack of Description for main model, there is no need of table 2, table 3, table 4 and Figure 2. 3.4, 3.5 section should be in start of material methods.

Author Response

(The authors gave the same response as above.)

Reviewer 3 Report

TITLE: A NEW STRATEGY FOR BEARING HEALTH ASSESSMENT WITH DYNAMIC INTERVAL PREDICTION MODEL

 COMMENTS AND QUESTIONS TO THE AUTHORS.

The subject of this paper is relevant and that should be of interest to Sensors mdpi readers. However, the revision is very difficult because of the lines in your manuscript are not numbered. In any case, I find very relevant mistakes of format (format references, description of the embedded formula, you should be consistent indicating in the same way “Table X, Figure X…” not table or figure with lower case, upper case or abbreviation randomly). The methodology is deficient, it should be better structured, I don't know how you obtained some graphics, or if the devices are well calibrated. In my honest opinion, I would like to review it again with the line numbers and with some issues (major and minor comments) that I indicate that I already see deficient.

Why do you use the technique of simulated annealing algorithm? I would like you to describe it at a general level and compare it with other optimization algorithms to justify this choice.

MAJOR COMMENTS

All the format of the references needs to be revised, now it's very confusing and prone to mistakes.

[Figure 1a] This Figure 1a is not clear, I do not see the units on the axes.

[Figure 1b] How did you get that Figure 1b? Through some software? From any bibliographical source? The units are very confusing because there is no explanation of them in the manuscript.

[Section 2.1] Why this definition of the health stage?

[Section 2.1] The motor characteristics of Figure 1b appear suddenly in the manuscript. I should have indicated them before, this apparition does not demonstrate scientific solidity.

[Section 2.1] Why this sampling interval?

[Section 2.1] I suggest that you include a graphic diagram of the described process (similar to Figure 2)

[Section 2.3] Why this formula 12? Please include bibliographic source if applicable.

[Section 3.1] Why these operating conditions?

[Figure 3] It is not clear for me how you obtain this Figure 3. Which device is used?

MINOR COMMENTS

In the second paragraph of the introduction, a full stop is missing.

Please delete the last paragraph of the introduction because it is unnecessary and also repetitive, the structure is seen in the manuscript.

[Figure 8] Actural value? This a mistake…(Actual value).

ENGLISH LANGUAGE AND STYLE

Moderate editing of English language required.

OVERALL RECOMMENDATION

Reconsider after major revision.

Moderate editing of English language required.

Author Response

(The authors gave the same response as above.)

Reviewer 4 Report

The manuscript presents a model for predicting the remaining useful lifetime of a bearing, based on mathematical and physical approaches. The method was validated using data from a long-term laboratory experiment, yielding interesting results. 

In general, the manuscript is well-written and the results are effectively presented. However, I recommend a minor English language proofread and checking for some formatting issues (especially citations).

Author Response

(The authors gave the same response as above.)

Round 2

Reviewer 3 Report

All my comments and suggestions have been included in this version, as well as the comments and suggestions of other reviewers. The present version has been significantly improved resulting in an interesting paper.

Minor editing of English language required.